# Preliminary Investigation of Mixed Orchard Hays on the Meat Quality, Fatty Acid Profile, and Gastrointestinal Microbiota in Goat Kids

**DOI:** 10.3390/ani12060780

**Published:** 2022-03-19

**Authors:** Yingying Wang, Tengfei Li, Xinyi Chen, Chongyi Liu, Xumei Jin, Hua Tan, Mingxiu Long

**Affiliations:** 1College of Grassland Agriculture, Northwest A&F University, Xianyang 712100, China; wyy2021@nwafu.edu.cn (Y.W.); lzu_litf@lzu.edu.cn (T.L.); c1971412778@163.com (X.C.); lcy27unlimited@gmail.com (C.L.); jinxumei@nwafu.edu.cn (X.J.); epiphyllumTH0305@163.com (H.T.); 2State Key Laboratory of Grassland Agro-Ecosystems, Key Laboratory of Grassland Livestock Industry Innovation, Ministry of Agriculture and Rural Affairs, College of Pastoral Agriculture Science and Technology, Lanzhou University, Lanzhou 730020, China

**Keywords:** ruminants, fatty acids, rumen bacteria, intestinal bacteria, association analysis

## Abstract

**Simple Summary:**

Goat meat constitutes one of the main animal protein sources in the human diet in developing countries. Consumers realize that there are multiple links between diet and happiness. Livestock feeding strategies can increase the concentration of a number of healthy fatty acids. Forages play an important role in maintaining rumen function. This preliminary investigation evaluated the effects of feeding different mixed hays grown in the orchard on meat quality, fatty acids, amino acids, and the gastrointestinal tract microbial ecosystem. The results indicated that goats fed alfalfa + oats mixed hay displayed a more favorable fatty acid profile and microbiota for human health. Feeding higher nutritional value forage is a good strategy for providing high-quality goat meat and is beneficial to the ruminant production system.

**Abstract:**

This preliminary investigation was designed to study the effects of different mixed orchard hays on meat quality, fatty acids, amino acids, rumen intestinal microflora, and the relationship between rumen bacteria and fatty acids in the *longissimus dorsi* muscle of Saanen dairy goats. In this preliminary investigation, goats were separately fed crop straws (corn and wheat straws) and alfalfa (*Medicago sativa* L.) (CK group), alfalfa + oats (*Avena sativa* L.) (group I), alfalfa + perennial ryegrass (*Lolium perenne* L.) (group II), and hairy vetch (*Vicia villosa* Roth.) + perennial ryegrass (group III). There were differences in shear force and cooking loss between treatments. The contents of saturated fatty acids (SFAs) C14:0, C16:0, and C18:0 in the CK group were significantly higher than those in other three groups (*p* < 0.001). The 16S rDNA sequencing results showed that the relative abundance of Proteobacteria in group II were higher than those in other three groups (*p* < 0.05). Association analysis showed that Prevotella_1 was negatively correlated with C18:0 and significantly positively correlated with C16:1, while Clostridium and Romboutsia showed a positive correlation with monounsaturated fatty acids (MUFAs) and polyunsaturated fatty acids (PUFAs). Therefore, feeding mixed hays can increase beneficial fatty acids and the percentages of associated bacteria in rumen and intestines.

## 1. Introduction

Roughage is an important part of feed for ruminants and other herbivores, and affects the growth and development of ruminants and the quality of ruminant products. In recent years, as more and more people focus on the relationship between nutrition and health, consumers are increasingly demanding healthier meat products [1]. However, there is a problem of inadequate high-quality diets for ruminants. The sustainable use of grassland and the supply of high-quality forage grass are the primary issues [2]. The “fruit–grass–livestock” composite system combines the fruit industry with the breeding industry, which is in line with the overall requirements of sustainable development and ecological planting and breeding. The hay produced by this system is a high-quality forage source for ruminants. Significant increases in ruminant production and productivity are needed to meet the growing demand for meat products.

Ruminants fed on pasture produce meat with higher omega-3 polyunsaturated fatty acids and conjugated linoleic acid, and superior nutritional value [3]. Compared with concentrate-based systems, forage-based production is more suitable for goats because it has a lower fat content [4]. The composition of plant secondary compounds that regulate ruminal biohydrogenation may largely differ between herbage species [5]. Forage quality and diet composition play important roles in the fatty acid profiles of meat from ruminants [6], and fatty acids are closely linked to human health. Food of ruminant origin is an important source of unsaturated fatty acids (UFAs) for the human body. A large proportion of the saturated fatty acids (SFAs) in the human diet comes from meat, which increases the risk of cardiovascular diseases [7]. Humans’ demand for healthy food increases continuously, requiring the intake of fewer SFAs and more UFAs [8]. Polyunsaturated fatty acids (PUFAs) in ruminant products depend on the availability of their precursors in the diet and the extent of biohydrogenation in the rumen [9].

The rumen and intestinal bacterial community is related to the diet. Ruminal microbes degrade carbohydrates to volatile fatty acids (VFAs), providing 70–80% of the metabolizable energy. The effects of different types of mixed hay on the rumen and intestinal bacterial community composition of Saanen dairy goats are still unclear. Little is known about the relationship between rumen or the intestinal bacterial community and the fatty acid profiles in the *longissimus dorsi* muscle of goats. Therefore, this preliminary investigation was conducted to study the content of fatty acids and amino acids in the *longissimus dorsi* muscle of goats, the effects of different mixed hays on rumen and intestinal bacterial populations by 16S rDNA sequencing, and the relationship between rumen or the intestinal bacteria and the fatty acid profiles of Saanen dairy goats. This will provide further data on how diet regulates rumen or the intestinal bacterial community, and reveal the dominant bacteria related to fatty acids in the *longissimus dorsi* muscle of Saanen dairy goats.

## 2. Materials and Methods

### 2.1. Experimental Animals and Design

The experiment was conducted on the Hai Yuan family farm in Pu Cheng County, Wei Nan City. A total of 24 healthy 2-month-old Saanen dairy male goats with an initial bodyweight of 11.48 ± 0.55 kg were randomly divided into four groups (according to the principle of similar body weight). There were 3 replicates per treatment and 2 goats per replicate. The experiment lasted for 85 days (the prefeeding period was 15 days and the trial period was 70 days). Single-stall feeding was adopted in the experiment. Goats in the control group were fed mixed roughage (mainly corn straw, wheat straw, alfalfa) prepared by local farmers; the other three experimental groups were respectively fed alfalfa + oats, alfalfa + perennial ryegrass, and hairy vetch + perennial ryegrass mixed hay. The nutrient level of mixed hays in each group was shown in Table 1.

### 2.2. Experimental Animal Feeding and Management

The experimental diet was formulated according to the Feeding Standard of Mutton Sheep (NY/T 816-2004). Before feeding, the forage grass in each group was cut short (3–5 cm in length) and mixed in a ratio of 1:1 to form mixed hay. The composition of concentrate supplement was: corn 68%, bran 3%, bean pulp 24%, baking soda 1%, and premix 4%. The premix (content per kilogram of feed) contained: vitamin A 2640 IU, vitamin D 340IU, vitamin E 26 mg, Fe 60 mg, Cu 12 mg, Zn 48 mg, Mn 48 mg, Co 0.12 mg, I 0.3 mg, and Se 0.36 mg. The concentrate feed was 0.8% of a goat’s body weight. Concentrate and roughage were fed separately to ensure that there was a 10% surplus in the manger. The four groups had the same concentrate-to-forage ratio.

Before the experiment, the shed was cleaned and disinfected, and epidemic prevention and internal deinsectization were carried out for all the experimental goats. During the experiment, goats were fed at 08:30 and 16:30 every day, and we ensured that there was still about 10% left in the feed tank before feeding on the morning of the second day. During the test, the goats were free to eat and drink water. The shed was cleaned, dewormed, regularly disinfected with a 3% NaOH solution, and ventilated.

### 2.3. Sample Collection

At the end of the trial period, 3 goats with similar body weights in each group were slaughtered commercially. Rumen and cecum contents were collected immediately after slaughter and stored in 5 mL cryostorage tubes at −80 °C for the determination of rumen and intestinal bacterial microflora. The *longissimus dorsi* muscle samples were collected between the 12th and 13th ribs from both the right and left sides of the carcass. Samples on the right were used to determine pH, color, drip loss, cooking loss, and shear force, while samples on the left were used to determine chemical composition. 

### 2.4. Meat Quality Determination

#### 2.4.1. Meat pH Determination

The meat pH values were measured at approximately 45 min and 24 h postmortem using a portable meat pH meter with a sharp penetrating blade over the electrode, as described by the American Meat Science Association. 

#### 2.4.2. Meat Color Determination

The *longissimus dorsi* muscle samples were exposed to air for approximately 45 min for color determination. A spectrophotometer was set to the L* (lightness), a* (redness), and b* (yellowness) system using illuminant D65, an observer angle of 10°, and an aperture size of 5.0 mm. Values of L*, a*, and b* were recorded from three readings performed at different points on the surface of each sample, using a HunterLab MiniScan EZ spectrophotometer.

#### 2.4.3. Drip Loss and Cooking Loss Determination

Drip loss was expressed as follows: drip loss (%) = ((initial weight − final weight)/initial weight) * 100. The method of determining drip loss was described in [10]. Cooking loss was expressed as follows: cooking loss (%) = ((precooked weight − postcooked weight)/precooked) * 100. The method of measuring cooking loss was described in [11].

#### 2.4.4. Shear Force Determination

The samples were cooked and refrigerated overnight, then six strips with a cross-sectional area of 1 cm^2^ were removed from each sample. A cutting blade was then used to measure the peak force to cut each sample strip perpendicular to the muscle fibers. The shear force was calculated as the average of 6 subsamples [11].

### 2.5. Fatty Acid Profile Analysis

Lipids from meat samples (5 g) were extracted by homogenizing in chloroform:methanol (2:1), shaken for 2 h, soaked for 8 h, and filtered with a G3 funnel. Then, 5 ml of 20% sodium chloride solution was added into the filtrate and statically stratified, and the lower layer of chloroform was the fat extract. After dehydration by anhydrous sodium sulfate, the fat was concentrated by rotary evaporation at 40 °C. Then, 5 mL of NaOH and methanol solution (0.5 mol/L) was added and refluxed at 70 °C for 5 min, followed by 5 mL of boron trifluoride ether solution at 70 °C for 2 min for fat methylation. Finally, 2 mL of chromatographic pure hexane was added. After reflux for 1 min at 70 °C, 5 mL of saturated NaCl solution was added and left to stand for 10 min. Then, 1 mL of n-hexane G was absorbed into the sample bottle and filtered with a 0.22 mm organic filter membrane for the gas chromatograph analysis [12]. Fatty acids were calculated as the sum of the FA profiles of GC quantification. All FA concentrations were expressed in mg/100 g of meat.

### 2.6. The Profiles of Total Amino Acids and Free Amino Acids

Total amino acids following hydrolysis and free amino acids (FAA) in deproteinized muscle were measured according to [13].

### 2.7. 16S rDNA Determination of Rumen and Intestinal Microorganisms

Total bacterial DNA was extracted with the E.Z.N.A.^®^ Stool DNA Kit, the extracted DNA quality was detected by agarose gel electrophoresis, and the DNA was quantified with a UV spectrophotometer. For PCR amplification of the 16S rDNA (V3 + V4) variable region, primers 341F (5′-CCTACGGGNGGCWGCAg-3′) and 805R (5′-GactachVGGGTATctaATCC-3′) were used. The PCR amplification procedures and steps were consistent with the study by Li et al., and were completed by Hangzhou Lian Chuan Biotechnology Co., LTD. (Hangzhou, China).

### 2.8. Statistical Analysis

The data were analyzed using a randomized complete block design with four different mixed hay types for each block. The experimental data consisted of three replicates per forage mixture group, for a total of 12 observations of the mixed hay type treatments. The meat quality characteristics, fatty acids, amino acids, rumen and intestinal bacteria alpha diversity, and composition data were analyzed using SPSS 23.0 software (SPSS Inc., Chicago, IL, USA). Alpha diversity metrics were calculated using the “diversity” function in the vegan package of R software version 3.5.2 (Lucent Technologies, Murray Hill, NJ, USA). One-way analysis of variance (ANOVA) was used to examine the changes among different groups. In addition, one-way ANOVA of the bacteria data was based on the normal distribution test. The data analysis method adopted in the ANOVA was the least significant difference (LSD) method. The results were expressed as the mean ± standard deviation (mean ± SD); *p* < 0.05 indicated a statistically significant difference, while *p* < 0.01 meant a very statistically significant difference. We used a Pearson correlation coefficient to assess the association between the rumen and intestinal microflora and fatty acids. For species annotation, sequence comparison was carried out with the feature-classifier plugin of QIIME2. The comparison database was the SILVA and NT-16S database, and the annotation result of the SILVA database was the criterion. Other graphs were drawn using Origin 2018.

## 3. Results

### 3.1. Final Body Weight

The final body weight of Saanen dairy goats fed different mixed hays was shown in Table 2.

### 3.2. Meat Quality Characteristics 

As shown in Table 3, the shear force was significantly lower in group II compared to CK group (*p* < 0.05), cooking loss was significantly determined by dietary treatments (*p* < 0.05), and meat sampled from goats in group II had higher cooking loss compared to the CK group and group III. The four treatment groups showed no significant effect (*p* > 0.05) on the drip loss percentage. Regarding the meat color, L* at 45 min in group II was recorded as higher (*p* < 0.05), but no differences were observed between CK, group I, and group II. Regarding b* at 45 min, they were quite similar (*p* > 0.05) among the treatment groups. With the extension of postmortem time, the pH of muscles in all four groups showed a decreasing trend, and the pH at 24 h in the four groups showed no differences (*p* > 0.05).

### 3.3. Relative Percentage of Fatty Acids in Longissimus Dorsi Muscle

The mixed hay type had effects (*p* < 0.05) on fatty acids in the *longissimus dorsi* muscle (Table 4). Goats fed crop stalks (corn and wheat) + alfalfa had the highest percentage of palmitic (C16:0) and stearic (C18:0) acids in the *longissimus dorsi* muscle, and showed a significant difference (*p* < 0.05) compared with the other three treatments, whereas the alfalfa + oats treatment resulted in the lowest content of palmitic (C16:0) acid. A higher percentage of myristic (C14:0) acid in the alfalfa + perennial ryegrass treatments and hairy vetch + perennial ryegrass treatments was detected compared with the CK and alfalfa + oats treatments. In addition, the concentration of myristic (C14:0) acid in goats fed with the alfalfa + perennial ryegrass and hairy vetch + perennial ryegrass showed no significant differences. There was a tendency for the alfalfa + oats, alfalfa + perennial ryegrass, and hairy vetch + perennial ryegrass diets to increase muscle MUFAs and PUFAs compared to the CK diet. Oleic (C18:1) acid was present in greater concentrations in the goats fed with the alfalfa + oats compared to other treatments. The goats fed with hairy vetch + perennial ryegrass displayed the highest concentrations of palmitoleic (C16:1) acid among four treatments. Linoleic (C18:2) acid was the most prominent PUFA, while alfalfa + perennial ryegrass–fed goats showed higher linoleic (C18:2) acid and lower α-linolenic (C18:3) acid when compared to other dietary treatments. The levels of α-linolenic (C18:3) acid in the goats fed alfalfa + oats were higher compared to those in goats fed the other three diets.

### 3.4. Essential Amino Acid and Nonessential Amino Acid Contents in longissimus dorsi Muscles of Saanen Dairy Goats

The effects of mixed hay type on amino acid content and composition of the *longissimus dorsi* muscles of Saanen dairy goats are given in Table 5. We noticed that concentrations of all amino acids in group I were greater than those in other groups. Concentrations of essential amino acids and nonessential amino acids had the same tendency in four groups, and the order of highest to lowest was: I > II > III > CK. The supplement of alfalfa + oat mixed hay to the diet showed particularly increased essential amino acids and nonessential amino acids compared with the other mixed hays. Concentrations of leucine, serine, and glutamic acids in the four treatment groups showed significant differences (*p* < 0.001). The goats fed alfalfa + oat, alfalfa + perennial ryegrass, or hairy vetch + perennial ryegrass mixed hay showed no significantly differences in the concentrations of histidine, proline, glycine and cystine. In particular, differences were observed in the concentrations of leucine, phenylalanine, histidine, serine, glutamic acid, and proline among the CK group and group III. Compared to group II, alfalfa + oat supplementation markedly increased amino acid concentrations, except for histidine, proline, glycine, and cystine. Overall, concentrations of EAA and NEAA in goats fed alfalfa + oat, alfalfa + perennial ryegrass, and hairy vetch + perennial ryegrass mixed hays were higher than in those fed the crop stalks (corn and wheat) + alfalfa.

### 3.5. Rumen Bacteria Diversity of Goats Fed with Different Mixed Hays

According to Table 6, 1571, 1750, 1328, and 1596 OTUs were unique to groups CK, I, II, and III, respectively, with no significant difference. With regard to the alpha diversity, no differences in community richness or evenness were observed, since none of the Shannon, Simpson, or Chao 1 indices were different among the four treatment groups (*p* > 0.1). The coverage index in the four groups was close to 99%, indicating that the rumen microbes had high coverage. In an unweighted principal coordinate analysis (PCoA) for different samples based on UniFrac (Figure 1b), the principal coordinates 1 and 2 accounted for 16.69% and 13.66% of the total variation, respectively. The rumen samples were completely separated without crossover under feeding with four different mixed hays, and samples of the same group had obvious aggregation. The UPGMA clustering diagram based on the unweighted UniFrac distance (Figure 1a) showed that each group of samples had a good clustering situation. Thus, the sample similarity within the group was higher than that between the groups, and there were differences in rumen microbes of goats fed with different mixed hays.

Figure 1 showed that the effects of different mixed hays on Beta diversity of rumen bacteria. According to the results of species annotation, a total of 21 species of rumen bacteria were detected at the phylum level of the Saanen goats. The relative abundance of bacterial genera of the four diet groups was reported (Figure 2a). Bacteroidetes, Firmicutes, Fibrobacteres, Spirochaetes, and Proteobacteria were the relatively abundant bacteria in the rumen microorganisms. As shown in Figure 2, the composition of rumen bacteria was mainly Bacteroidetes, and the relative abundance of Bacteroidetes in group III was significantly higher than that in the other three groups (*p* < 0.05); however, the relative abundance of Firmicutes and Fibrobacteres was lower than that in the other three groups. The relative abundance of Proteobacteria in group II was higher than in the other three groups. At the genus level (Figure 2b), Prevotella_1, Rikenellaceae_RC9_gut_group, Fibrobacter, and Christensenellaceae_R7_group were the dominant bacterium groups. The relative abundance of Prevotella in group III was significantly higher than that in other groups (*p* < 0.05); Compared with the control group, groups II and III showed a higher relative abundance of the Rikenellaceae_RC9_gut_group (*p* > 0.05), whereas the relative abundance of Fibrobacter showed the opposite trend, and was lower in groups II and III compared to the CK group. 

### 3.6. Intestinal Bacteria Diversity of Goats Fed with Different Mixed Hays

Table 7 shows that the OTU, Shannon, and Chao 1 indices in group II were significantly higher than in the other groups (*p* < 0.01), and the Simpson index of intestinal bacteria in the orchard mixed hay group was significantly higher than that in control group (*p* < 0.01). The coverage index in the four groups was close to 99%, indicating that the intestinal microbes had high coverage. 

Figure 3 showed that the effects of different mixed hays on Beta diversity of intestinal bacteria. According to the results of species annotation, a total of 27 species of intestinal bacteria were detected at the phylum level of the Saanen goats (Figure 4a). Firmicutes, Proteobacteria, Actinobacteria, Cyanobacteria, and Bacteroidetes were the relatively abundant bacteria in the intestinal microorganisms. The relative abundance of Firmicutes in groups I, II, and III were significantly higher than in the CK group (*p* < 0.05). The relative abundance of Actinobacteria in the intestines were significantly higher (*p* < 0.05) in goats fed the crop stalks (corn and wheat) + alfalfa than in any other treatments. Group II showed a particularly higher (*p* < 0.05) relative abundance of Bacteroidetes compared to other groups. At the level of genus (Figure 4b), Romboutsia, Lawsonia, Clostridium, Christensenellaceae_R7_group, and Ruminococcaceae_NK4A214_group had higher counts compared with the others. The relative abundance of Romboutsia and Clostridium in groups I, II, and III were significantly higher (*p* < 0.05) than in the CK group; on the contrary, the relative abundance of Lawsonia was significantly lower (*p* < 0.05) than in the CK group. The relative abundances of the Christensenellaceae_R7_group in group II and the CK group were higher (*p* < 0.05) than in groups I and III.

### 3.7. Correlation Analysis of Rumen Microflora and Fatty Acids of Saanen Dairy Goats

Pearson’s correlations between fatty acids and rumen bacterial composition at the genus level are shown in Figure 5. C14:0 and C16:0 were negatively associated with F082_unclassified, Fibrobacter, Christensenellaceae_R-7_group, and Lachnospiraceae_XPB1014_group; C14:0 also correlated positively with Prevotellaceae_UCG-001 and Rikenellaceae_RC9_gut_group. C18:0 was the only saturated fatty acid negatively correlated with Muribaculaceae_unclassified. C16:1 correlated positively with Muribaculaceae_unclassified and Prevotella_1, whereas Prevotella_1 only had a significant relationship with C16:1. The concentrations of C18:1 and C18:3 were positively correlated with the genus Muribaculaceae_unclassified and F082_unclassified. C18:2 was positively associated with the relative abundances of Prevotellaceae_UCG-001 and Rikenellaceae_RC9_gut_group. 

### 3.8. Correlation Analysis of Intestinal Microflora and Fatty Acids of Saanen Dairy Goats

The fatty acids that showed high and significant (*p* < 0.05) Pearson’s correlations with intestinal bacterial microbiome composition at the genus level are reported in Figure 6. C14:0 showed a negative correlation with Firmicutes_unclassified and Saccharofermentans. Lawsonia was the only genus that showed a positive correlation with C16:0, whereas Romboutsia, Firmicutes_unclassified, Saccharofermentans, and Ruminococcus_1 were negatively correlated with C16:0. Ruminococcaceae_UCG-014, Clostridium, and Romboutsia showed a negative correlation with C18:0; on the contrary, positive correlations were observed between Ruminococcaceae_UCG-014, Clostridium, Romboutsia, and C16:1. At the same time, Christensenellaceae_R-7_group and Lawsonia showed positive correlations with C18:0, but a negative correlation with C16:1. Negative correlations were also found for C18:1 (Christensenellaceae_R-7_group), C18:2 (Lawsonia), and C18:3 (Christensenellaceae_R-7_group and Lawsonia). C18:1 showed a positive correlation with Clostridium, Firmicutes_unclassified, and Saccharofermentans. C18:2 was the only fatty acid positively correlated with Ruminococcaceae_NK4A214_group and Ruminococcus_1; in addition, C18:3 showed positive correlations with Clostridium, Romboutsia Firmicutes_unclassified, and Saccharofermentans.

## 4. Discussion

### 4.1. Meat Quality Characteristics

Meat color is an important factor affecting consumers’ purchases, and the main factors affecting the color of meat are pH and intramuscular fat [14]. Values of lightness (L*) below 34 and redness (a*) below 9.5 are considered dark and unacceptable to consumers [15]. Thus, the mean values for redness in groups CK and II were not acceptable. The water-holding capacity of meat is an important feature regarding meat quality, mainly regarding the loss of water during cooking of the meat, and tenderness is identified as the trait most associated with consumer acceptance [16]. In our study, the shear force value observed in group II was higher than that in other groups; however, the cooking loss in group II was lower. These results were contradictory, possible because the level of fat in the meat affected its tenderness, and with a higher level of fat, less cutting force was required [17]. With regard to meat pH at 24 h, there were no differences between groups. The pH 24 h value in this study was close to the recommended pH value for good meat quality of small ruminants, ranging from 5.5 to 5.8 [18].

### 4.2. Relative Percentage of Fatty Acids in Longissimus Dorsi Muscles

Ruminant meat fat generally presents high SFAs and low PUFAs due to the biohydrogenation of UFAs in feed in the rumen [12]. Palmitic and stearic acids are biohydrogenation products from the linolenic acid in forages. These are the most abundant FAs [7]. In this study, alfalfa + oat mixed hay had the lowest SFA content throughout the experiment, because oat is rich in phenolic compounds. Legumes decrease the extent of biohydrogenation, which has been attributed to the high activity of polyphenol oxidase in legumes, which inhibits lipolysis [19]. It is also possible that the difference in neutral detergent fiber (NDF) content between the mixed hays played an important role in reducing biohydrogenation of forage PUFAs [7]. The goats fed with crop stalks (corn and wheat) + alfalfa and alfalfa + perennial ryegrass mixed hay had lower monounsaturated fatty acid (MUFA) contents, which can be explained by the greater vitamin E content of alfalfa compared to other diets. High levels of vitamin E may accelerate the biohydrogenation of C18:1 UFA in the rumen [20]. 

The present study showed alfalfa + oat and alfalfa + perennial ryegrass mixed hays increased the concentration of linoleic (C18:2) acid. Therefore, the alfalfa + oat and alfalfa + perennial ryegrass mixed hays could be an option to improve the nutritional quality of goat meat, because linoleic acid (C18:2)-rich food prevents cardiovascular disease and hypertension in humans [21]. The concentration of alpha-linolenic acids (C18:3 omega-3 ALA) was higher in the *longissimus dorsi* muscles of goats fed alfalfa + oat than for the other forage types, which suggested that ALA in alfalfa + oat was better digested and absorbed by the goats [12]. Linoleic acid (C18:2) and alpha-linolenic acids (C18:3) are respectively metabolized to arachidonic acid (C20:4n-6) and EPA (eicosapentaenoic acid 20:5n-3). ALA is a precursor of DHA [22]. EPA and DHA are the main omega-3 fatty acids. The current study observed that meat from all treatments exceeded the EPA + DHA level of 26 mg/100 g of meat, which is the lowest EPA + DHA level, and is considered the lower limit for claiming goat meat as a “source” of omega-3 fatty acids [23].

### 4.3. Essential Amino Acid and Nonessential Amino Acid Contents of longissimus dorsi Muscles in Saanen Dairy Goats

The types and concentrations of amino acids are considered key factors to determine the nutritional value and flavor of meat [24], and play an important role in animal and human health [25]. Thus, we investigated the effects of different mixed hays on the compositions and concentrations of amino acids in the *longissimus dorsi* muscles of Saanen dairy goats. In the present study, the concentrations of all amino acids fed alfalfa + oat mixed hay were the highest throughout the experiment. Alfalfa hay and oat hay have good nutritional complementarity. A large number of amino acids were produced after degradation in the rumen. Alfalfa is rich in protein, mineral elements, vitamins, and bioactive substances such as saponins and flavonoids [26]. Alfalfa leaf protein contains 18 amino acids, including eight essential amino acids especially rich in lysine. The lysine content of the *longissimus dorsi* muscle was the highest in essential amino acids, and mixed hay containing alfalfa had a relatively higher lysine content than the other diets. 

The amino acids absorbed by ruminants are derived from microbial protein synthesis in the rumen, and from dietary amino acids undegraded in the rumen [27]. Due to the existence of rumen, amino acids in the diets of ruminants can be degraded by rumen microorganisms, resulting in the types and quantities of amino acids provided by diets that cannot reflect the absorption and utilization levels of amino acids in ruminants [28]. Avenanthramides and saponins are characteristically found in oat [29]. Saponins have an inhibitory effect on protozoa in rumen microorganisms, reduce the degradation of amino acids in rumen, and increase the number of amino acids reaching the small intestine and improve ruminant performance [30]. Therefore, the positive combination effect of alfalfa + oat effectively improved the digestibility of roughage and animal performance. 

### 4.4. The Rumen Microflora Composition of Goats Fed with Mixed Hay

In the present study, dietary type did not affect the alpha diversity of the rumen microbial community. The rumen of ruminants contains diverse and dense microorganisms that are essential for the bioconversion of feed that cannot be digested by the host’s digestive system [31]. This study indicated that Bacteroidetes and Firmicutes were the most abundant microbial phyla in the rumen flora, and Fibrobacteres were the third most abundant microbial phylum. The phyla Bacteroidetes and Firmicutes could degrade the complex plant polysaccharides [32]. A previous study showed that the composition and abundance of rumen microorganisms were affected by various factors, such as dietary structure and the ratio of fineness to roughness [33]. The relative abundance of Bacteroidetes in the alfalfa + oat mixed hay decreased significantly (*p* < 0.05), while the relative abundance of Firmicutes was significantly higher (*p* < 0.05). The effects of feeding different roughages on the rumen microbes of goats were studied. It was found that high-fiber diets led to Fibrobacteres showing an increasing trend [34]. As one of the major cellulose-degrading bacteria in the rumen, Fibrobacteres may play an important role in degrading low-quality forages [35]. A previous study showed that the relative abundance of Fibrobacteres was positively associated with the content of NDF [36]. In the present study, the NDF content of the crop stalks (corn and wheat) was the highest. Thus, the relative abundance of Fibrobacteres in the rumen of goats fed crop stalks (corn and wheat) + alfalfa was higher than in the other three groups. 

Prevotella_1 constitutes a major dominating genus and is more abundant in high-fiber diets, and plays an important role in the rumen ecosystem [37]. According to a previous study, Prevotella_1 abundance was positively correlated with the crude protein content, and alfalfa hay contains more crude protein compared with other hays, so a higher abundance of Prevotella_1 was observed in alfalfa hay [36]. However, the results of this research were inconsistent with previous results, which could have been caused by an inadequate amount of alfalfa in the mixed hay. The present study revealed higher abundances of unclassified F082 and unclassified Muribaculaceae. The function of these microorganisms is unknown, and their abundances were affected by the hay types. The possible explanation might be related to the potential roles of these organisms in the degradation of fiber. In the current study, the abundances of these two unclassified genera were higher in the rumen of goats fed alfalfa + oat, alfalfa + perennial ryegrass, and hairy vetch + perennial ryegrass mixed hays compared with the CK group. Thus, they were positively correlated with acid detergent fiber content.

### 4.5. The Intestinal Microflora Composition of Goats Fed with Mixed Hay

As the posterior digestive tract of ruminants, the intestinal tract is an important place for digestion and absorption of nutrients, and in which colonized microorganisms play an important role, helping ruminants acquire nutrients from their feed [38]. The cecum is the second major fermentation site in ruminants. The current research showed that the intestinal microbiota diversity and bacterial species richness were significantly higher in group II than those in the CK group. The results indicated that the high-quality forage diet could increase intestinal microbiota diversity. Firmicutes, Proteobacteria, Actinobacteria, and Bacteroidetes were the predominant phyla, with Firmicutes being the most abundant within each group except for the CK group. Consistent with previous findings, these dominant phyla are of ecological and functional importance in the ruminant intestinal tract. Postruminal degradation of cellulose and starch occurs in the cecum, which is crucial in ruminant digestion [39]. The abundance and diversity of microorganisms in the cecum were significantly higher than those in the rumen [40,41]. Firmicutes plays an important role in digesting fiber and cellulose [42]. Firmicutes comprises amounts of Gram-positive bacteria, and most of them are considered as beneficial bacteria for maintaining the balance of gut microbiota and preventing the invasion of pathogens [43]. Polyphenols can repress Firmicutes, favoring Bacteroidetes in the gut [44]. In this study, the relative abundance of Bacteroidetes in the cecum of goats fed alfalfa + perennial ryegrass mixed hay was the highest. The possible reason was that the perennial ryegrass hay contained more polyphenols. Bacteroidetes are responsible for the digestion of carbohydrates and proteins, and contribute to the maturation of the intestinal immune system. The results showed that the abundance of Firmicutes and Bacteroidetes in the cecum environment may be closely related to the energy and nutrient requirements of animals in their early growth and development [45]. The phylum Proteobacteria was abundant in cecum bacterial populations investigated in the goats fed crop stalks (corn and wheat) + alfalfa and hairy vetch + perennial ryegrass in the present study, which could have been due to their importance in polysaccharide degradation [46]. However, the relative abundance of Proteobacteria was very low in the other two groups. This difference was observed due to differences in the diet composition, because the constituent of the gut microbial community is related to the diet type [47]. Additionally, Proteobacteria exhibit a high diversity of metabolic functions that contribute to the high energy and nutritional requirements of a host [42]. 

### 4.6. Correlation Analysis of Rumen Microflora and Fatty Acids of Saanen Dairy Goats

The fatty acids in ruminant products are mainly affected by dietary nutrition and rumen metabolism of bacteria [48]. Rumen microbial fermentation plays an important role in impacting the fatty acid composition of meat by providing precursors for de novo fatty acid synthesis [49]. Therefore, a broad variety of fatty acids in ruminant products are mainly derived from the ruminal metabolism of their fat, rather than from the ruminant diet [50]. A previous study reported that 68–84% of UFAs in the diet were converted to SFAs by bacterial biohydrogenation [51]. Because biohydrogenation of dietary fatty acids is usually incomplete, many intermediate metabolites reaching the duodenum can be incorporated into tissues [49]. Manipulating the microbial lipid metabolism can promote the outflow of beneficial fatty acids from the rumen [3]. One study found dietary strategies that facilitated reasonable fatty acids compositions in ruminant products via regulation of the bacterial community [52]. Ruminal lipid metabolism is a key to determine the ideal fatty acid content in ruminant products [3]. In the present study, Prevotella_1 was negatively correlated with C18:0 and significantly positively correlated with C16:1, which was in accordance with Purushe’s research, in which it was found that Prevotella had a significantly positive correlation to PUFA content, but was negatively related to SFA content. Prevotella plays a vital role in energy collection in the rumen ecosystem, and provides precursors for UFA synthesis [53]. In this study, the genera Muribaculaceae_unclassified and F082_unclassified could affect fatty acids, and directly or indirectly promoted PUFA synthesis and inhibited SFA synthesis, which indicated that they restrained biohydrogenation translating of PUFA to SFA in the rumen. In addition, the genera Fibrobacter, Christensenellaceae_R-7_group, and Lachnospiraceae_XPB1014_group were correlated negatively with C14:0 and C16:0 proportions. However, abundances of Lawsonia and Christensenellaceae_R-7_group were positively related to C16:0 and C18:0 contents. These bacterial genera are essential for biohydrogenation in the cecum. Clostridium and Romboutsia showed a positive correlation with MUFAs and PUFAs, as Clostridium production can oxidize to provide energy for ruminants. 

## 5. Conclusions

This preliminary investigation demonstrated different mixed hay types affecting the rumen and cecum microflora of goats using 16S rDNA gene sequencing. It showed that alfalfa + oats and alfalfa + perennial ryegrass mixed hays had better effects, and alfalfa + oats mixed hay significantly increased beneficial fatty acids and amino acids in the *longissimus dorsi* muscles. Different mixed hay feeding of Saanen dairy goats could improve the diversity of rumen and intestinal bacterial community. The association analysis showed a strong correlation between rumen or intestinal bacteria and fatty acids in the *longissimus dorsi* muscles, and revealed the specific bacteria associated with fatty acids. However, further research is required with a larger sample of animals to confirm the results of this study.

## Figures and Tables

**Figure 1 animals-12-00780-f001:**
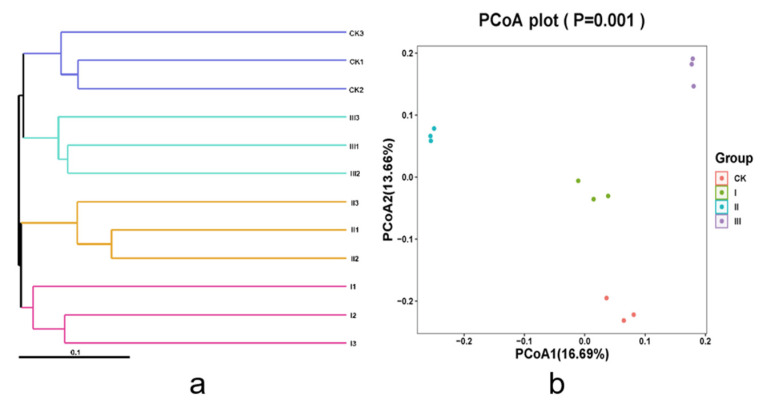
Principal coordinate analysis (PCoA) (**b**) based on the unweighted UniFrac distance (**a**) of the rumen microbial communities.

**Figure 2 animals-12-00780-f002:**
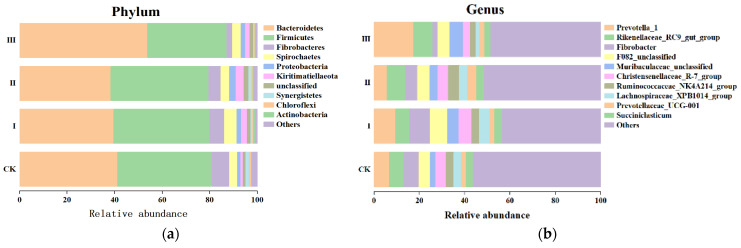
Composition of rumen bacterial communities in goats at phylum (**a**) and genus (**b**) levels.

**Figure 3 animals-12-00780-f003:**
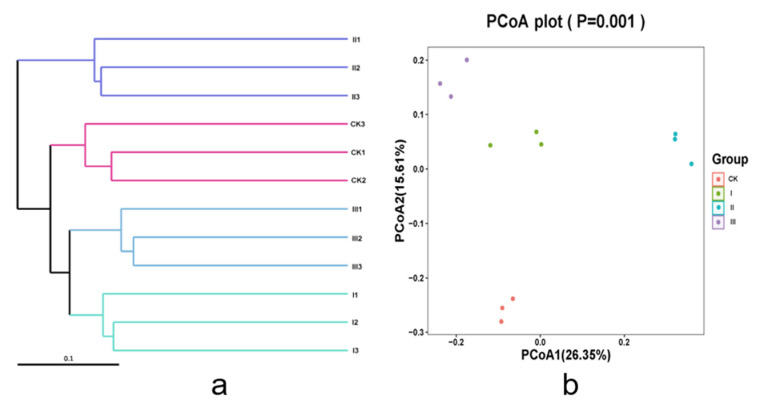
Principal coordinate analysis (PCoA) (**b**) based on the unweighted UniFrac distance (**a**) of the intestinal microbial communities.

**Figure 4 animals-12-00780-f004:**
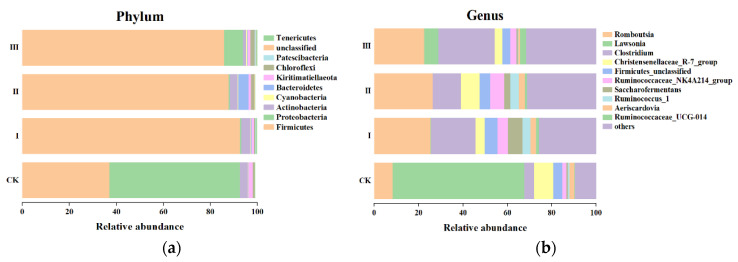
Composition of intestinal bacterial communities in goats at phylum (**a**) and genus levels (**b**).

**Figure 5 animals-12-00780-f005:**
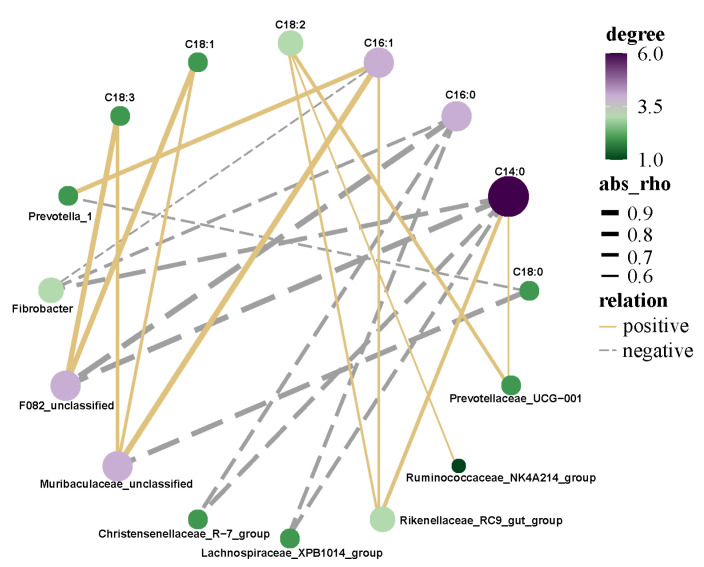
Pearson’s correlations between rumen microflora and fatty acids.

**Figure 6 animals-12-00780-f006:**
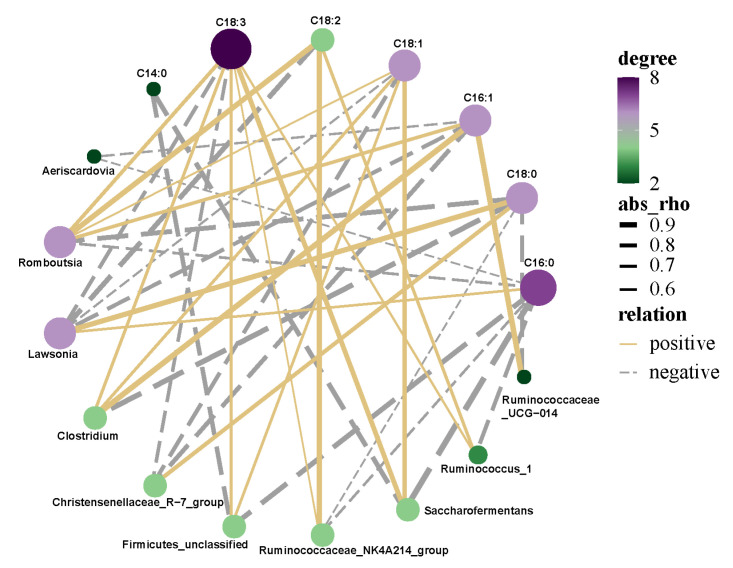
Pearson’s correlation between intestinal microflora and fatty acids.

**Table 1 animals-12-00780-t001:** Nutrient level of mixed hay in each group (DM basis)%.

Items	Group			
	CK ^1^	I ^2^	II ^3^	III ^4^
Dry matter (%)	81.28	84.90	85.98	83.72
Crude protein (%)	13.32	15.5	16.80	16.00
Neutral detergent fibre (%)	41.57	40.90	41.20	35.70
Acid detergent fibre (%)	26.50	28.5	27.70	29.30
Crude ash (%)	11.90	10.80	11.53	12.20

^1^ CK—crop stalks (corn and wheat) + alfalfa; ^2^ I—alfalfa + oats; ^3^ II—alfalfa + perennial ryegrass; ^4^ III—hairy vetch + perennial ryegrass.

**Table 2 animals-12-00780-t002:** Final body weight of Saanen dairy goats fed different mixed hays.

Group	Final Body Weight/kg
CK ^1^	16.99 ± 0.72 ^c^
I ^2^	21.92 ± 0.84 ^a^
II ^3^	19.9 ± 0.74 ^b^
III ^4^	19.73 ± 0.21 ^b^

Different letters (a, b, c) within a row indicate statistically significant differences (*p* < 0.05). ^1^ CK –crop stalks (corn and wheat) + alfalfa; ^2^ I—alfalfa + oats; ^3^ II—alfalfa + perennial ryegrass; ^4^ III—hairy vetch + perennial ryegrass.

**Table 3 animals-12-00780-t003:** Meat quality characteristics of Saanen dairy goats fed different mixed hays.

Items	Group				*p*-Value	SEM ^5^
	CK ^1^	I ^2^	II ^3^	III ^4^		
Shear force/N	37.45 ± 0.32 ^a^	35.76 ± 1.15 ^a,b^	33.22 ± 1.50 ^b^	37.87 ± 0.17 ^a^	0.03	1.36
Cooking loss/%	62.53 ± 1.33 ^c^	65.71 ± 0.66 ^a,b^	66.17 ± 0.39 ^a^	63.25 ± 0.40 ^b,c^	0.03	1.12
Drip loss/%	3.43 ± 0.01	3.30 ± 0.003	2.99 ± 0.35	3.26 ± 0.01	0.41	0.25
L* 45 min	36.81 ± 0.45 ^a,b^	35.35 ± 0.63 ^b^	37.97 ± 0.85 ^a^	36.35 ± 0.35 ^a,b^	0.08	0.85
a* 45 min	9.46 ± 0.68 ^a,b^	10.48 ± 0.46 ^a^	8.56 ± 0.28 ^b^	10.51 ± 0.09 ^a^	0.04	0.62
b* 45 min	11.29 ± 0.56	11.03 ± 0.57	11.17 ± 1.19	11.77 ± 0.46	0.91	1.07
pH 45 min	6.69 ± 0.01 ^a,b^	6.66 ± 0.01 ^b^	6.69 ± 0.01 ^a^	6.66 ± 0.01 ^b^	0.07	0.01
pH 24 h	5.62 ± 0.01	5.61 ± 0.003	5.62 ± 0.01	5.62 ± 0.01	0.80	0.01

Different letters (a, b, c) within a row indicate statistically significant differences (*p* < 0.05). ^1^ CK—crop stalks (corn and wheat) + alfalfa; ^2^ I—alfalfa + oats; ^3^ II—alfalfa + perennial ryegrass; ^4^ III—hairy vetch + perennial ryegrass; ^5^ SEM—standard error of mean.

**Table 4 animals-12-00780-t004:** Effect of mixed hay from different orchards on fatty acid content and composition (% of total fatty acids) of the *longissimus dorsi* muscle in Saanen dairy goats.

Fatty Acid (%)	Group				*p*-Value	SEM
	CK ^1^	I ^2^	II ^3^	III ^4^		
C16:0	19.66 ± 0.06 ^a^	15.63 ± 0.15 ^d^	18.13 ± 0.09 ^c^	19.16 ± 0.09 ^b^	<0.001	0.14
C18:0	28.90 ± 0.23 ^a^	23.73 ± 0.12 ^c^	24.83 ± 0.39 ^b^	23.17 ± 0.19 ^c^	<0.001	0.36
C14:0	1.26 ± 0.02 ^b^	0.81 ± 0.00 ^c^	1.38 ± 0.04 ^a^	1.38 ± 0.01 ^a^	<0.001	0.32
C16:1	0.56 ± 0.05 ^d^	1.18 ± 0.01 ^b^	1.05 ± 0.02 ^c^	1.58 ± 0.01 ^a^	<0.001	0.39
C18:1	44.70 ± 0.26 ^c^	50.14 ± 0.03 ^a^	45.20 ± 0.45 ^c^	46.97 ± 0.23 ^b^	<0.001	0.41
C18:2	3.59 ± 0.30 ^c^	6.14 ± 0.05 ^b^	7.72 ± 0.08 ^a^	5.92 ± 0.03 ^b^	<0.001	0.22
C18:3	1.24 ± 0.02 ^d^	2.36 ± 0.02 ^a^	1.65 ± 0.01 ^c^	1.75 ± 0.01 ^b^	<0.001	0.02
n-6: n-3	2.89 ± 0.21 ^c^	2.60 ± 0.03 ^c^	4.68 ± 0.05 ^a^	3.39 ± 0.03 ^b^	<0.001	0.15

Different letters (a, b, c, d) within a row indicate a very statistically significant difference (*p < 0.001*). ^1^ CK—crop stalks (corn and wheat) + alfalfa; ^2^ I—alfalfa + oats; ^3^ II—alfalfa + perennial ryegrass; ^4^ III—hairy vetch + perennial ryegrass; ^5^ SEM—standard error of mean.

**Table 5 animals-12-00780-t005:** Effects of mixed hays from different orchards on amino acid content and composition (g/100 g) of the *longissimus dorsi* muscle in Saanen dairy goats.

Amino Acid (g/100 g)	Group				*p*-Value	SEM
	CK	I	II	III		
Total amino acids	19.55 ± 0.46 ^d^	24.43±0.41^a^	22.13±0.11^b^	20.72±0.22 ^c^	<0.001	0.47
Essential amino acids						
Threonine	0.94 ± 0.02 ^c^	1.18 ± 0.01 ^a^	1.06 ± 0.01 ^b^	0.98 ± 0.01 ^c^	<0.001	0.02
Valine	0.91 ± 0.02 ^c^	1.17 ± 0.01 ^a^	1.03 ± 0.01 ^b^	0.95 ± 0.02 ^c^	<0.001	0.02
Methionine	0.51 ± 0.01 ^b^	0.63 ± 0.01 ^a^	0.55 ± 0.02 ^b^	0.52 ± 0.02 ^b^	0.002	0.02
Isoleucine	0.89 ± 0.02 ^c^	1.12 ± 0.02 ^a^	0.99 ± 0.01 ^b^	0.92 ± 0.01 ^c^	<0.001	0.02
Leucine	1.58 ± 0.03 ^d^	1.98 ± 0.03 ^a^	1.77 ± 0.01 ^b^	1.69 ± 0.01 ^c^	<0.001	0.03
Tyrosine	0.67 ± 0.02 ^c^	0.85 ± 0.01 ^a^	0.76 ± 0.01 ^b^	0.71 ± 0.01 ^c^	<0.001	0.02
Phenylalanine	0.85 ± 0.02 ^c^	1.10 ± 0.01 ^a^	0.96 ± 0.02 ^b^	0.93 ± 0.02 ^b^	<0.001	0.02
Lysine	1.86 ± 0.04 ^c^	2.32 ± 0.03 ^a^	2.06 ± 0.04 ^b^	1.89 ± 0.06 ^c^	<0.001	0.06
Histidine	0.70 ± 0.02 ^b^	0.84 ± 0.01 ^a^	0.83 ± 0.00 ^a^	0.81 ± 0.03 ^a^	0.002	0.03
Nonessential amino acids						
Asparagine	1.87 ± 0.04 ^c^	2.35 ± 0.02 ^a^	2.09 ± 0.02 ^b^	1.96 ± 0.03 ^c^	<0.001	0.04
Serine	0.76 ± 0.01 ^d^	0.95 ± 0.01 ^a^	0.87 ± 0.01 ^b^	0.81 ± 0.01 ^c^	<0.001	0.01
Glutamic acid	2.93 ± 0.07 ^d^	3.64 ± 0.06 ^a^	3.29 ± 0.03 ^b^	3.10 ± 0.02 ^c^	<0.001	0.07
Proline	1.59 ± 0.11 ^c^	2.12 ± 0.09 ^a^	2.02 ± 0.02 ^a,b^	1.86 ± 0.04 ^b^	0.004	0.1
Glycine	0.93 ± 0.01 ^b^	1.05 ± 0.05 ^a^	1.00 ± 0.02^a,b^	0.94 ± 0.01 ^b^	0.078	0.04
Alanine	1.22 ± 0.02 ^c^	1.48 ± 0.05 ^a^	1.34 ± 0.02 ^b^	1.28 ± 0.02 ^b,c^	0.001	0.04
Cystine	0.13 ± 0.00 ^b^	0.16 ± 0.01 ^a^	0.14 ± 0.01 ^a,b^	0.13 ± 0.01 ^a,b^	0.082	0.01
Arginine	1.22 ± 0.03 ^c^	1.47 ± 0.04 ^a^	1.38 ± 0.01 ^b^	1.24 ± 0.02 ^c^	0.001	0.04

Different letters (a, b, c, d) within a row indicate statistically significant differences (*p* < 0.05) and very significant differences (*p* < 0.001).

**Table 6 animals-12-00780-t006:** Alpha-diversity indices in the rumen microbiota of Saanen dairy goats fed different mixed hays.

Items	Group				*p*-Value	SEM
	CK	I	II	III		
OTUs	1571 ± 55	1750 ± 134	1328 ± 48	1596 ± 148	0.12	151
Shannon	9.02 ± 0.09	9.10 ± 0.14	9.11 ± 0.08	9.09 ± 0.11	0.94	0.15
Simpson	0.99 ± 0.003	0.99 ± 0.003	1.00	0.99 ± 0.003	0.33	0.004
Chao1	1592.05 ± 59.22	1782.02 ± 148.10	1331.70 ± 47.11	1618.25 ± 159.94	0.12	163.16
Goods_coverage	0.997	0.996	0.999	0.997	0.19	0.001

**Table 7 animals-12-00780-t007:** Alpha-diversity indices in the intestinal microbiota of Saanen dairy goats fed different mixed hays.

Items	Group				*p*-Value	SEM
	CK	I	II	III		
OTU	581 ± 22 ^b,c^	728 ± 116 ^b^	1025 ± 75 ^a^	449 ± 53 ^c^	<0.01	106
Shannon	3.79 ± 0.27 ^d^	6.31 ± 0.19 ^b^	7.04 ± 0.14 ^a^	5.28 ± 0.23 ^c^	<0.01	0.30
Simpson	0.65 ± 0.08 ^b^	0.95 ± 0.01 ^a^	0.95 ± 0.01 ^a^	0.92 ± 0.01 ^a^	<0.01	0.03
Chao1	597.57 ± 24.92 ^b,c^	735.61 ± 119.72 ^b^	1040.46 ± 88.90 ^a^	451.93 ± 54.46 ^c^	<0.01	113.63
Goods_coverage	0.998	0.999	0.997	0.999	0.11	0.001

Different letters (a, b, c, d) within a row indicate statistically significant differences (*p* < 0.05).

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
