# Peer review of "Preliminary Investigation of Mixed Orchard Hays on the Meat Quality, Fatty Acid Profile, and Gastrointestinal Microbiota in Goat Kids"

_animals, 2022, doi:10.3390/ani12060780_

Round 1

Reviewer 1 Report

Animals Manuscript ID 1594190

Meat Quality, Fatty Acid Profile and Microbiota across Gastrointestinal Tract of Goats Subjected to Different Mixed Hay Grown in Orchard

Wang et al.

The objective of the study is valid; however, there are concerns about methodology, that will prevent acceptance of the manuscript in its present form.

Abstract

What is the rationale for using Saanen dairy goats for assessing meat quality and composition?

Why abbreviate the control group as CK group?

The total number of goats used in the experiment and in each treatment must be included.

All scientific names of organisms and muscle must be italicized.

Lines 28-29:  “There were no differences in meat quality characteristics between treatments.”

Line 33: “…was negatively correlated…”

Line 36:  “…and the percentages of associated bacteria in rumen and intestines.”

Introduction

Line 43:  “…demanding healthier meat products [1].”

Line 53: “…and conjugated linoleic acid…”

Lines 57-59:  Is this sentence true?  Please check.

Line 62:  PUFA – expand when using abbreviation for the first time.

Italicize Longissimus dorsi throughout the manuscript.

Line 64:  “…bacterial community…”

Line 74:  “…diet regulates rumen or intestinal…”

Materials and Methods

A total of 24 animals were used with 6 animals in each diet group. Out of the 6, only 3 animals were processed for sample collection.  This is not enough for the results to be statistically valid.

Lines 86-87:  Was the diet 50-50 in groups I, II, and III?

Section 2.2 and elsewhere: the methodology must be written in past tense.

Line 100:  “…goats were fed…”

Lines 102-103:  not clear; please elaborate.

Sample Collection

The animals would have been approx. 5 month old at slaughter and sample collection.  Given the fact these were dairy goat kids with small L. dorsi muscles and the quantities of samples collected were small (between 12th and 13th rib), it is not clear how pH, color, shear force (mentioned in line 113, but no methodology given), drip loss, cooking loss, fatty acids, and amino acids were analyzed.

The biggest concern I have is the number of animals used to collect muscle samples. 

Author Response

Replies to Editor and Reviewer:

We would like to express our sincere thanks to your relevant comments and constructive suggestions. These comments are very helpful for revising and improving our paper, and are important guiding significance to our other researches. We have studied the comments carefully and made corrections which we hope to meet with approval. The point to point responses to the comments and suggestion are as follows:

Answers to the comments from Reviewer 1:

We are thankful for the reviewer’s careful reading and comments to improve the manuscript. In the following, we will respond to the comments one by one.

  1. What is the rationale for using Saanen dairy goats for assessing meat quality and composition?

Answer: Our experiment was conducted in Shaanxi province, China, Saanen dairy goats are the main local breed, we use Saanen dairy goats for assessing meat quality and composition, the research results can be better used in production practice.

  1. Why abbreviate the control group as CK group?

Answer: In general, we abbreviate the control check as CK group.

  1. The total number of goats used in the experiment and in each treatment must be included.

Answer: Twenty-four healthy 2-month-old Saanen dairy goats with an initial bodyweight of 11.48±0.55 kg were randomly divided into four treatments, and there were 3 replicates per treatment and 2 goats per replicate, so, there were 6 goats in each treatment during feeding.

  1. All scientific names of organisms and muscle must be italicized.

Answer: Done, we accepted the suggestion and have revised it all in the article.

  1. Lines 28-29: “There were no differences in meat quality characteristics between treatments.”

Answer: Done, we revise the sentence and highlight it in red in the text.

  1. Line 33: “…was negatively correlated…”

Answer: Done, we revise the sentence and highlight it in red in the text.

  1. Line 36: “…and the percentages of associated bacteria in rumen and intestines.”

Answer: Done, we revise the sentence and highlight it in red in the text.

  1. Line 43: “…demanding healthier meat products [1].”

Answer: Done, we revise the sentence and highlight it in red in the text.

  1. Line 53: “…and conjugated linoleic acid…”

Answer: Done, we revise the sentence and highlight it in red in the text.

  1. Lines 57-59: Is this sentence true? Please check.

Answer: We check this sentence, and revise the sentence and highlight it in red in the text.

  1. Line 62: PUFA – expand when using abbreviation for the first time.

Answer: Done, we use PUFA-expand polyunsaturated fatty acids for the first time and highlight it in red in the text.

  1. Italicize Longissimus dorsi throughout the manuscript.

Answer: Done, we italicized Longissimus dorsi throughout the manuscript, and highlight it in red in the text.

  1. Line 64: “…bacterial community…”

Answer: Done, we revise the sentence and highlight it in red in the text.

  1. Line 74: “…diet regulates rumen or intestinal…”

Answer: Done, we revise the sentence and highlight it in red in the text.

  1. A total of 24 animals were used with 6 animals in each diet group. Out of the 6, only 3 animals were processed for sample collection. This is not enough for the results to be statistically valid.

Answer: To avoid errors, we deleted 3 animals with large individual differences, and chose 3 animals with the similar weight to analyze in each group. The weight of each group was CK (16.99±0.72 kg), group I (21.92±0.84 kg), group II (19.9±0.74 kg), group III (19.73±0.21 kg).

  1. Lines 86-87: Was the diet 50-50 in groups I, II, and III?

Answer: Yes, the diet was 50-50 in groups I, II, and III.

  1. Section 2.2 and elsewhere: the methodology must be written in past tense.

Answer: Done, we accepted the suggestion and have revised it.

  1. Line 100: “…goats were fed…”

Answer: Done, we revise the sentence and highlight it in red in the text.

  1. Lines 102-103: not clear; please elaborate.

Answer: Done, we elaborated this sentence and highlight it in red in the text.

  1. Sample Collection

Answer: We supplement the shear force measurement method, and highlight it in red in the text.

Overall, we thank the reviewer and editor again for their constructive comments and beneficial suggestions. We have made some significant improvements and revised some problems in the revised manuscript. We hope that the reviewer can be satisfied with our responses to concerns and the corresponding revisions of the manuscript. We sincerely hope that this revised manuscript can be accepted for publication in Animals.

Reviewer 2 Report

Article`s review

As follows from the article`s content,the aim was to study  influence of various types of hay on the quality of meat, the content of fatty acids, amino acids, the state of the microflora of the rumen and caecum, as well as identifying the relationship between the microbiome of the rumen and the formation of fatty acids and amino acids in the longest back muscle of Zaanen goats. The conducted studies are relevant because they are aimed at identifying the relationships between the state of the microflora of the rumen and the caecum and the amount of polyunsaturated fatty acids in the longest back muscle, the content of essential and interchangeable amino acids, functional and technological characteristics of meat.
The authors have shown that by selecting the feeding diet of goats, it is possible to control the formation of the desired microflora in the rumen and intestines purposefully. It has been established that by feeding with certain types of hay Zaanen dairy goats, it is possible to significantly improve the diversity of the bacterial community of the rumen and intestines.
At the same time, it was found that the use of mixed hay: alfalfa + perennial ryegrass gives the best effect, and mixed hay: alfalfa + oats significantly increases the content of useful fatty acids and amino acids in the longest back muscle.
The research methodology does not raise any questions, the results obtained are interpreted scientifically.

Comments on the work:
1. It is unclear from the materials of the article how many goats (males and females) participated in the experiment. This may affect the results of research.
2. In our opinion, the authors conducted an insufficient sample of animals for the experiment.
3. Tables 2 and 3 indicate the reliability of the difference in results between the compared groups, but in many cases they are not calculated accurately.
4. The note to Table 3 is made incorrectly and requires more detailed interpretation (since only P≤0.05 is indicated, and there are P≤0.01 and 0.001).
5. the list of references in the article has 8 references for the last 5 years, and the rest are earlier. It is advisable to cite more recent literary sources.

After such a minor correction, the article can be recommended for publication.

Author Response

Replies to Editor and Reviewer:

We would like to express our sincere thanks to your relevant comments and constructive suggestions. These comments are very helpful for revising and improving our paper, and are important guiding significance to our other researches. We have studied the comments carefully and made corrections which we hope to meet with approval. The point to point responses to the comments and suggestion are as follows:

Answers to the comments from Reviewer 2:

We are thankful for the reviewer’s careful reading and comments to improve the manuscript. In the following, we will respond to the comments one by one.

  1. It is unclear from the materials of the article how many goats (males and females) participated in the experiment. This may affect the results of research.

Answer: A total of 24 Saanen dairy male goats were used with 6 goats in each diet group, and there were 3 replicates per treatment and 2 goats per replicate. To avoid errors, we deleted 3 goats with large individual differences, and chose 3 goats with the same weight to analyze in each group.

  1. In our opinion, the authors conducted an insufficient sample of animals for the experiment.

Answer: To avoid errors, we deleted 3 goats with large individual differences, and chose 3 goats with the similar weight to analyze in each group. The weight of each group was CK (16.99±0.72 kg), group I (21.92±0.84 kg), group II (19.9±0.74 kg), group III (19.73±0.21 kg).

  1. Tables 2 and 3 indicate the reliability of the difference in results between the compared groups, but in many cases they are not calculated accurately.

Answer: The data processing is consistent with our results, and we refer to the following two literatures: [1, 2]

  1. Santos-Silva, J., Francisco, A., Alves, S.P., Portugal, P., Dentinho, T., Almeida, J., Soldado, D., Jeronimo, E. and Bessa, R.J.B. Effect of dietary neutral detergent fibre source on lambs growth, meat quality and biohydrogenation intermediates. Meat Science 2019, 147, 28-36.
  2. Fruet, A.P.B., Trombetta, F., Stefanello, F.S., Speroni, C.S., Donadel, J.Z., De Souza, A.N.M., Rosado Junior, A., Tonetto, C.J., Wagner, R., De Mello, A. and Nornberg, J.L. Effects of feeding legume-grass pasture and different concentrate levels on fatty acid profile, volatile compounds, and off-flavor of the M. longissimus thoracis. Meat Science 2018, 140, 112-118.

  1. The note to Table 3 is made incorrectly and requires more detailed interpretation (since only P≤0.05 is indicated, and there are P≤0.01 and 0.001).

Answer: Done, we corrected the note and highlight it in red in the text.

  1. The list of references in the article has 8 references for the last 5 years, and the rest are earlier. It is advisable to cite more recent literary sources.

Answer: Done, we have replaced some of the older literature with recent literary sources, and the rest of the literature cannot be replaced.

Overall, we thank the reviewer and editor again for their constructive comments and beneficial suggestions. We have made some significant improvements and revised some problems in the revised manuscript. We hope that the reviewer can be satisfied with our responses to concerns and the corresponding revisions of the manuscript. We sincerely hope that this revised manuscript can be accepted for publication in Animals.

Reviewer 3 Report

Did you make sample size analysis before conducting the study, 3 replicates per treatments seems too limited to have conclusions.

Experimental design needs clarification. Where the two goats per replicate in the same pen?

Do you have any performance data? Body weight (you mentioned you selected goats with similar BW to slaughter? Intake?

In the abstract is mentioned that there are no meat quality characteristics, and there are some in Table 2.

Could you better describe the statistic section? It surprises me how you can find significant difference and nice figures using only 3 animals per treatments in meat quality and bacteria profile

Author Response

Replies to Editor and Reviewer:

We would like to express our sincere thanks to your relevant comments and constructive suggestions. These comments are very helpful for revising and improving our paper, and are important guiding significance to our other researches. We have studied the comments carefully and made corrections which we hope to meet with approval. The point to point responses to the comments and suggestion are as follows:

Answers to the comments from Reviewer 3:

We are thankful for the reviewer’s careful reading and comments to improve the manuscript. In the following, we will respond to the comments one by one.

  1. Did you make sample size analysis before conducting the study, 3 replicates per treatments seems too limited to have conclusions.

Answer: Before conducting the study, we analyzed the sample size. To avoid errors, we deleted 3 goats with large individual differences, and chose 3 goats with the same weight to analyze in each group. The weight of each group was CK (16.99±0.72 kg), group I (21.92±0.84 kg), group II (19.9±0.74 kg), group III (19.73±0.21 kg).

  1. Experimental design needs clarification. Where the two goats per replicate in the same pen?

Answer: We chose 24 goats of the similar weight, so two goats had similar weight which were randomly assigned to the same pen.

  1. In the abstract is mentioned that there are no meat quality characteristics, and there are some in Table 2.

Answer: Done, we added meat quality characteristics to the abstract, and highlight it in red in the text.

  1. Could you better describe the statistic section? It surprises me how you can find significant difference and nice figures using only 3 animals per treatments in meat quality and bacteria profile.

Answer: Done, we added a more detailed statistical analysis and highlight it in red in the text. About data, first, we can guarantee the reliability of the data. Second, we chose 3 goats with similar weight from 6 goats in each treatment, in order to reduce the error and ensure the validity of data. Third, we measured and averaged the data of 3 goats, and then conducted statistical analysis with SPSS 23.0 software.

Round 2

Reviewer 1 Report

The authors have sincerely attempted to address the comments made in round 1; however, the main issue of sample size has not been addressed satisfactorily.  I still feel 3 animals per treatment is not adequate for the results to be reliable.  I strongly recommend including all 6 animals in the analysis to increase power and revising the manuscript accordingly.  It is likely the results will change when all six animals are included, but those results will be more reliable than what is presented now. 

Author Response

Reply to Editor and Reviewer:

We would like to express our sincere thanks to your relevant comments and constructive suggestions. These comments are very helpful for revising and improving our paper, and have important guiding significance to our other researches. We have studied the comments carefully and made responses which we hope to meet with approval.

Answers to the comments from Reviewer 1:

The authors have sincerely attempted to address the comments made in round 1; however, the main issue of sample size has not been addressed satisfactorily. I still feel 3 animals per treatment is not adequate for the results to be reliable. I strongly recommend including all 6 animals in the analysis to increase power and revising the manuscript accordingly. It is likely the results will change when all six animals are included, but those results will be more reliable than what is presented now.

Answer: First of all, I would like to elaborate on the sample size of our experiment: twenty-four healthy 2-month-old Saanen dairy male goats with an initial bodyweight of 11.48±0.55 kg were randomly divided into four treatments, and there were 3 replicates per treatment and 2 goats per replicate, so, there were 6 goats in each treatment during feeding. To avoid errors, we deleted animals with large individual differences, and chose 3 animals with the similar weight to analyze in each group. The weight of each group was CK (16.99±0.72 kg), group I (21.92±0.84 kg), group II (19.9±0.74 kg), group III (19.73±0.21 kg). Secondly, since we have completed the whole experiment, and we’re sorry we cannot resample now. Thirdly, I referred to two articles published in Animals that also used three replicates of each treatment and found reliable results.[1, 2].

Overall, we thank the reviewer and editor again for their constructive comments and beneficial suggestions. We hope that the reviewer can be satisfied with our responses to concerns. We sincerely hope that this revised manuscript can be accepted for publication in Animals.

  1. An, X., Zhang, L., Luo, J., Zhao, S. and Jiao, T. Effects of Oat Hay Content in Diets on Nutrient Metabolism and the Rumen Microflora in Sheep. Animals 2020, 10.
  2. Ramos, S.C., Jeong, C.D., Mamuad, L.L., Kim, S.H., Kang, S.H., Kim, E.T., Cho, Y.I., Lee, S.S. and Lee, S.S. Diet Transition from High-Forage to High-Concentrate Alters Rumen Bacterial Community Composition, Epithelial Transcriptomes and Ruminal Fermentation Parameters in Dairy Cows. Animals 2021, 11.

Round 3

Reviewer 1 Report

I feel this can be published as a preliminary study.  Toward the end of the Conclusion section, please mention that further research is required with a larger sample of animals to confirm the results of this study... 

Author Response

Dear Reviewer, thanks to your relevant comments and constructive suggestions, we have studied the comments carefully and mentioned that further research is required with a larger sample of animals to confirm the results of this study at the end of the Conclusion section. We hope that the reviewer can be satisfied with our responses to concerns. We sincerely hope that this revised manuscript can be accepted for publication in Animals.
